# Infectious Complications in Home Parenteral Nutrition: A Systematic Review and Meta-Analysis Comparing Peripherally-Inserted Central Catheters with Other Central Catheters

**DOI:** 10.3390/nu11092083

**Published:** 2019-09-04

**Authors:** Raquel Mateo-Lobo, Javier Riveiro, Belén Vega-Piñero, José I. Botella-Carretero

**Affiliations:** 1Department of Endocrinology and Nutrition, Hospital Universitario Ramón y Cajal, 28034 Madrid, Spain; 2Centro de Investigación Biomédica en Red Fisiopatología de la Obesidad y la Nutrición (CIBEROBN), 28034 Madrid, Spain

**Keywords:** home parenteral nutrition, central catheter, peripherally inserted central catheter, port, tunneled catheter, catheter-related infection

## Abstract

Background: Home parenteral nutrition (HPN) has become a common therapy. There is still controversy regarding the possibility that peripherally inserted central catheters (PICCs) may diminish catheter-related blood stream infection (CRBSI) rates. Methods: We searched the PubMed database for studies reporting the rates of CRBSI with HPN. Study selection was performed independently by three investigators. Disagreements were discussed and resolved by consensus or by arbitration by an author not involved in the search. The National Institutes of Health Quality Assessment Tools was used to assess the methodological quality of the studies. Meta-analyses were performed using MetaXL 5.3 with the quality effects model. Results: Screening of the article titles and abstracts yielded 134 full text articles for evaluation. Only three prospective studies that included appropriate data were considered for the final analysis. The relative risk of the CRBSI rate was 0.41 (0.14–1.17) for PICC vs. tunneled catheters. The relative risk of the CRBSI rate was 0.16 (0.04–0.64) for PICC vs. ports. The relative risk of the thrombosis rate was 3.16 (0.20–49.67) for PICCs vs. tunneled. Conclusions: There is insufficient evidence to show a difference in CRBSI rates between PICCs and tunneled catheters. On the other hand, PICCs showed lower CRBSI rates than ports. There was also no difference in the rate of catheter-related thrombosis and mechanical complications.

## 1. Introduction

In the last few years, parenteral nutrition has become a common therapy for patients at home and has been mainly driven by an increase in the proportion of adult patients with cancer who need this therapy [1,2]. National registries in North America, including more than a thousand patients, showed that nearly 30% were expected to require home parenteral nutrition (HPN) indefinitely [3]. In Spain, a total of 308 patients were registered in 2017 from 45 centers with 3012 episodes, which represent a prevalence rate of 6.61 patients/million inhabitants/year, which is higher than in previous reports [1]. Furthermore, as prognosis and survival in the long term are better for adults with HPN than for those with an intestinal transplant, the latter is reserved for those presenting severe complications of parenteral nutrition [4]. As more patients are progressively included in HPN programs, the standardization of care and the development of good education programs will be a cornerstone in contributing to an improvement in results, with special emphasis on caregivers [5]. 

The most frequent of the severe complications in HPN is a catheter-related blood stream infection (CRBSI), affecting more than 10% of patients with this therapy. CRBSI per 1,000 parenteral nutrition-days ranges from 0.35 to 0.91 in adults [6]. Those with ports or double-lumen catheters showed more infections than those with peripherally-inserted central catheters (PICCs), tunneled-central catheters, or single-lumen catheters in the North America National registry [6]. This result has also been recently confirmed by a prospective study of our group [7].

Although European guidelines for the choice of central venous access for HPN state that a tunneled central catheter is preferred over other types of central catheters [8], in the last Canadian registry [2], the proportion of PICCs increased from 21.6% to 52.9% [2]. Therefore, there is still some controversy regarding the possibility that PICC use may diminish CRBSI rate compared to other central venous catheters (CVCs). Some studies showed that patients with a PICC for HPN had an increased rate of infections [9,10]. On the other hand, a recent meta-analysis [11] has shown a lower rate of CRBSI in HPN patients using PICCs compared with tunneled central venous catheters. However, the authors of this meta-analysis found that single-arm studies showed that the rate of CRBSI in PICC and the rate of other central catheters were comparable. Furthermore, most of the included studies in the aforementioned meta-analysis were retrospective ones. Owing to these controversies, and in the view of the publication of some recent studies, we found it interesting to perform a systematic review and meta-analysis of the incidence of CRBSI in HPN, focusing on the use of PICC vs. other CVC in prospective studies.

## 2. Methods

We followed the MOOSE Guidelines for Meta-Analyses and Systematic Reviews of Observational Studies [12] and the Preferred Reporting Items for Systematic Reviews and Meta-Analyses recommendations [13]. The present meta-analysis was registered in Prospero (CRD42019127130).

### 2.1. Data Sources and Searches

We searched the PubMed online facilities, introducing the following as MESH terms: (("parenteral nutrition, home"[MeSH Terms] OR ("parenteral"[All Fields] AND "nutrition"[All Fields] AND "home"[All Fields]) OR "home parenteral nutrition"[All Fields] OR ("home"[All Fields] AND "parenteral"[All Fields] AND "nutrition"[All Fields])) OR ("parenteral nutrition, home total"[MeSH Terms] OR ("parenteral"[All Fields] AND "nutrition"[All Fields] AND "home"[All Fields] AND "total"[All Fields]) OR "home total parenteral nutrition"[All Fields] OR ("home"[All Fields] AND "total"[All Fields] AND "parenteral"[All Fields] AND "nutrition"[All Fields])) OR (home[All Fields] AND ("parenteral nutrition"[MeSH Terms] OR ("parenteral"[All Fields] AND "nutrition"[All Fields]) OR "parenteral nutrition"[All Fields] OR "parenteral"[All Fields]))) AND (("infection"[MeSH Terms] OR "infection"[All Fields]) OR (("blood circulation"[MeSH Terms] OR ("blood"[All Fields] AND "circulation"[All Fields]) OR "blood circulation"[All Fields] OR "bloodstream"[All Fields]) AND ("infection"[MeSH Terms] OR "infection"[All Fields])) OR ("catheter-related infections"[MeSH Terms] OR ("catheter-related"[All Fields] AND "infections"[All Fields]) OR "catheter-related infections"[All Fields] OR ("catheter"[All Fields] AND "related"[All Fields] AND "infection"[All Fields]) OR "catheter related infection"[All Fields]) OR (catheter-related[All Fields] AND ("sepsis"[MeSH Terms] OR "sepsis"[All Fields])) OR (catheter-related[All Fields] AND ("blood circulation"[MeSH Terms] OR ("blood"[All Fields] AND "circulation"[All Fields]) OR "blood circulation"[All Fields] OR "bloodstream"[All Fields]) AND ("infection"[MeSH Terms] OR "infection"[All Fields]))). The last access was 30 April 2019, as stated in the registered protocol for the screening phase completion.

### 2.2. Study Selection

The following criteria were used to include studies in this meta-analysis: (1) English language articles; (2) prospective observational studies or clinical trials (retrospective studies or cases-control studies were not eligible); (3) the study must report the type of catheter studied (PICC vs. other CVCs); (4) the study should report associated and adequate epidemiological data to allow further statistical analyses; (5) we excluded studies that recruited pediatric patients; as our unit is dedicated only to adult patients, we have no experience with HPN in children.

Study selection was performed independently by three investigators (R.M.-L., J.R., B.V.-P.). Disagreements were discussed and resolved by consensus or by arbitration by an author not involved in the search of the literature (J.I.B.-C.).

### 2.3. Data Extraction

The following data were extracted from the retrieved studies: author, year, country, type of study, sample size (n), mean age (and SD), female/males, oncologic/non-oncologic patients, catheter type, catheters number, catheter days, infection rate expressed as n/1000 days, catheter lumens (if available), cultures, care protocol, thrombosis episodes/1000 days, heparin/saline or antibiotic lock, and other complications (such as tip displacement/catheter removal). When these data were not included in the published articles or unavailable even after contacting the authors by email, they were recorded as not reported (NR).

### 2.4. Quality Assessment

The National Institutes of Health Quality Assessment Tools for Controlled Intervention Studies and for Observational Cohort and Cross-Sectional Studies were used to assess the methodological quality of the included studies (see Appendix A) [14].

### 2.5. Statistical Analysis

Meta-analyses were performed using the MetaXL 5.3 program (http://www.epigear.com/index_files/metaxl.html) with the quality effects model for meta-analyses [15]. The quality effects model, which relies on the use of the quality indexes described above to weight the studies, is more robust compared with the fixed- or random-effects models when analyzing heterogeneous studies.

The effect size measure was the relative risk (RR) of the complication rates expressed as the number of a specific complication per days of catheter use, to compare PICCs with other CVCs. Forest plots showed the estimates as diamonds, with their lateral points indicating confidence intervals. The left hand column included study identifiers together with Cochran’s Q and I^2^ heterogeneity statistics, whereas the right-hand columns included forest plots found in each of these studies (squares and horizontal lines representing confidence intervals) and their corresponding numerical information. The assessment of publication bias by funnel plots [16] and Doi plots with Luis Furuya–Kanamori (LFK) indexes of asymmetry [15] was only considered if 10 or more studies were analyzed. All P values were two-sided, and α = 0.05 was set as the level of statistical significance.

## 3. Results

Figure 1 shows the PRISMA flow chart, from the identification of studies to their inclusion in the meta-analysis. The search strategy yielded 755 studies for initial review. Screening of the article titles and abstracts yielded 134 full text articles for evaluation (reasons for the screen-out are given in Figure 1). After reading the selected articles, only three prospective studies that included appropriate data for both PICCs and other CVCs were considered for the final analysis [7,17,18]. We discarded eight articles with retrospective, pediatric, or inpatient data [9,10,19,20,21,22,23,24], two articles with duplicated series of patients [25,26], and seven articles with inappropriately reported infection rates for adequate extraction [27,28,29,30,31,32,33], whereas other reasons for the exclusion of full text articles (such as including several of the above reasons and others, such as different types of reported outcomes apart from catheter-related infections, the use of locks as preventive measures, the type of lipid emulsions, and other unrelated outcomes) are given in Figure 1 (references are listed in the Appendix A). 

### 3.1. Characteristics of Selected Studies and Patient Demographics

Table 1 shows the detailed characteristics of the included studies. All three studies were undertaken in Europe [7,17,18]. All of them were prospective observational cohorts, with no randomization for the allocation of catheters. These studies included from 151 to 254 patients for a total of 601 when combined. There was one study that included only oncologic patients [17], while the other two included both oncologic and non-oncologic patients [7,18]. All of studies included men and women and compared PICCs to other CVCs—for example, PICCs vs. Hickman tunneled catheters and ports [7], PICCs vs. Broviac tunneled catheters [18], and PICCs vs. Groshong tunneled catheters, ports, and non-tunneled catheters [17].

### 3.2. Meta-Analysis of CRBSI Rates with PICCs vs. Other Central Catheters

A total of 44,321 catheter days were observed for PICCs and 83,753 for the rest of catheters in the three included studies [7,17,18] (Table 2). The smallest CRBSI rates were observed for PICCs in all the studies, and the highest rates were observed with the ports in one study [7] and with non-tunneled central catheters in another one [17]. Non-tunneled central catheters were not included in the meta-analysis because only one study included them. Cultures that define CRBSI as qualitative blood cultures from a peripheral vein and from the catheter, or meet criteria for quantitative blood cultures or the differential time to positivity, were used in all the included studies (Table 2). Two studies reported specific care protocols for catheters with an aseptic technique [7,17]. In the other study, this information was not adequately reported [18] (Table 2). 

When meta-analyzing the CRBSI rates for PICCs vs. tunneled catheters, there were a total of 44,321 vs. 48,814 catheter days, respectively, with 15 CRBSI confirmed by cultures for PICCs and 75 for tunneled catheters, representing CRBSI rates of 0.15 [7], 1.05 [18], and 0 [17] for PICCs and 0.72 [7], 1.87 [18], and 0.64 [17] for tunneled catheters in the three included studies, respectively (Table 2). The RR of the CRBSI rate was 0.41 (0.14–1.17) for the PICCs vs. tunneled catheters, with a small to moderate heterogeneity (Figure 2).

When meta-analyzing the CRBSI rates for PICCs vs. ports, there were a total of 31,999 vs. 24,575 catheter days respectively, with three CRBSIs, confirmed by cultures for PICCs, and seven for ports, representing CRBSI rates of 0.15 [7], 0 [17] for PICCs, and 2.35 [7] and 0.19 [17] for ports in the two included studies (Table 2). The RR of the CRBSI rate was 0.16 (0.04–0.64) for PICCs vs. ports with a small heterogeneity (Figure 3).

We were not able to perform a meta-analysis of the CRBSI rate comparing single lumen vs. multilumen catheters, as in one of the three studies, only single lumen catheters were used [17], and in another one, these data were not available [18] (Table 2). 

### 3.3. Meta-Analysis of Non-infectious Catheter Related Complications

As PICCs and tunneled catheters showed no significant differences in their rates of infectious complications in the meta-analysis, we then compared the occurrence of other non-infectious complications between PICCs vs. tunneled catheters. 

The catheter insertion techniques differed from those in the three studies reported in Table 3. Flushing with heparin after the parenteral nutrition infusion was employed in one study [7], antibiotic lock was performed in another (taurolidine-citrate locks were injected in 35% of patients, for the rest the flushing compound was not reported in this study) [18], and saline was used in the other one [17] (Table 3). 

Catheter thrombosis in symptomatic patients after ultrasound diagnosis was reported only with PICCs with a rate of 0.049 per 1000 catheter days in one study [7], 0.4 in the second one [18], and none in the third one [17] (Table 3). The RR of the thrombosis rate was 3.16 (0.20–49.67) for PICCs vs. tunneled catheters with moderate heterogeneity (Figure 4).

The reported mechanical complications included catheter dislocation, catheter rupture, or catheter occlusion (Table 3). One study reported only one catheter tip displacement with a PICC [7], whereas mechanical complications were more common in the other two studies for both the PICCs and tunneled catheters (Table 3). The RR of the mechanical complications rate was 1.0 (0.52–1.92) for PICCs vs. tunneled catheters with a low heterogeneity (Figure 5).

## 4. Discussion

### 4.1. CRBSI Rates with PICCs vs. Other Central Catheters

In the present study we have shown that CRBSI rates were similar between PICCs and tunneled central catheters, but PICCs showed fewer infections compared to ports in prospective studies. Previous studies showed that PICCs had fewer infectious complications than other CVC when used for many intravenous therapies, as reported by Maki et al. [34], whose meta-analysis showed that the rate of CRBSI was 1.0 per 1000 catheter days for PICC and 1.6 per 1000 catheter days for tunneled catheters. A recent meta-analysis that focused specifically on the rate of CRBSI in patients with HPN showed a lower rate of CRBSI in HPN patients using PICCs compared with tunneled central catheters [11]. However, an analysis of single-arm studies in this same meta-analysis showed that the rate of CRBSI was comparable between both types of catheters. This meta-analysis included both prospective (*n* = 3) and retrospective studies (*n* = 2), with the consequent risk of bias modifying the results. 

Of the two retrospective studies included in the latter meta-analysis, one by Elfassy et al. [19] included 202 PICCs and 62 tunneled catheters reported a 1.96 and 1.93 CRBSI rates for these catheters, respectively, and the other by Christensen et al. [10] included 126 PICCs and 169 tunneled catheters and reported 1.63 and 0.56 CRBSI rates for these catheters, respectively. Patient selection was done by identifying all adult patients with intestinal failure requiring HPN by clinic charts and electronic medical records from 2001 to 2008 [19] or by local archives from 2008 to 2014 [10] in a retrospective way. In the study by Elfassy et al. [19], patient data were excluded from the analyses if the date of the line insertion was not documented or if the patients did not attend their clinic for a follow-up. In the study by Christensen et al. [10], records were included for all patients who were dependent on HPN at that time, patients with finalized treatment, and patients who were deceased. Therefore, these different inclusion criteria may have biased the results. In addition, although in the study by Christensen et al. [10] all catheters were single lumen ones, in the study by Elfassy et al. [19], the type was not reported. Moreover, patients in the former study needed a total of 295 catheters over a 6 y period, which shows a higher rate of catheter replacement than in other studies.

Two other recently published retrospective studies were not included in the meta-analysis by Hon et al. [11] and neither in our present study: one of them included 191 PICCs and only 11 tunneled central catheters with a 0.61 and 0.93 CRBSI rate per 1000 catheter days for these catheters respectively [21], which could have biased our result in favor of PICCs. The other study included patients identified through medical records during a predefined 3 year period, reporting 123 PICCs and 51 tunneled central catheters with a 1.78 and 1 CRBSI rate per 1000 catheter days for these catheters, respectively [9]. Although including these four recent published retrospective studies could have increased the statistical power of our meta-analysis, we considered that the risk of bias was high and thus decided to include only prospective studies.

The rate of CRBSI in the three prospective included studies did not differ after meta-analysis between PICCs and tunneled catheters. Although this is a consistent result with the data of the single arm studies of the previous published meta-analysis by Hon et al. [11], we were not able to analyze the impact of the number of lumens of the catheters on the CRBSI rate. In fact, we have recently reported that CRBSI per 1000 catheter-days showed no difference between the PICCs and Hickman, but the rates were more frequent with multilumen catheters [7]. This is also consistent with the results of Ross et al. in their recent report, in which patients with double-lumen catheters also had more CRBSI than those with PICCs or single-lumen catheters [6].

On the other hand, our meta-analysis demonstrates that PICCs have less CRBSI than ports, which is also consistent with our previous results [7] and also those reported by Ross et al. [6]. Port systems are totally implantable venous silicone or polyurethane catheters with subcutaneous reservoir chambers made of titanium or ceramic. The port membrane is made of silicone and is only punctured with special port cannulae (non-coring port needles) [8]. It is recommended that the port needle be replaced every third to seventh day in patients receiving HPN with cyclical nutritional application [35]. Therefore, the use of the needle and its more frequent manipulation may increase the CRBSI rate compared to PICCs and tunneled catheters.

### 4.2. Non-infectious Catheter Related Complications

Catheter-related central vein thrombosis is a severe complication of HPN and may be a concern while choosing a type of catheter for delivering parenteral nutrition, especially for the long term. Cuerda et al. [36] have recently reported a multicenter prospective study of sixty-two patients (31 males, 31 females) aged 50 ± 19 years, who followed-up for a median 363 days, for whom the study found an incidence of catheter-related thrombosis of 0.045 per catheter/year. In this study, the thrombosis rate was not significantly associated with any of the variables analyzed, including the type of catheter [36]. This is in agreement with the results of our present study, although there was a trend for more episodes of catheter-related thrombosis with PICCs than with tunneled catheters, this was not significant after the meta-analysis. 

In fact, of the three included studies in our meta-analysis, only one showed a significant incidence of greater thrombosis with PICCs (Toure et al. [18]). In this study, the specific care protocol of the catheters was not reported in detail (and the number of single lumen or multilumen catheters was also not reported), and taurolidine-citrate locks were injected in 35% of patients, but for the rest, the flushing compound was not reported. Furthermore, ultrasound was employed to diagnose catheter-related thrombosis only in patients presenting with symptoms like arm swelling, pain, loss of function, and head or neck swelling [18]. The different care protocols in the three studies may have produced differences in the rate of catheter-related thrombosis, as good catheter care requires an experienced multidisciplinary team trained to identify and aggressively treat catheter-related complications [36].

We found no differences in the mechanical complications between PICCs and other tunneled catheters, and this result was consistent and homogeneous for the three included studies in our meta-analysis. Therefore, the choice of the type of catheter for HPN need not rely on the risk of catheter-related thrombosis or other non-infectious complications.

### 4.3. Limitations

As we identified only three prospective studies after the systematic review for inclusion in the meta-analysis, there is a limitation in terms of statistical power. Further, the allocation of patients to one or another type of catheter was not randomized, and the selection of the catheters for each patient was performed on a clinical basis with a consequent risk of bias. In the study by Santacruz et al. [7], the choice of the CVC was not randomized but based on the patient´s responsible physician, always taking into account the underlying disease, the expected duration of HPN, and the possibility of a safe procedure for obtaining a venous access. In the study by Toure et al. [18], the choice of CVC was made by the patient’s physician; a PICC catheter was preferred for patients initially expected to have HPN for fewer than 6 months, and Broviac was preferred for those patients initially expected to need HPN for more than 6 months. In the study by Cotogni et al. [17], the choice of the type of catheter in each patient was largely based on the provider´s preference, since at the time of their study, there was no official hospital policy or recommendation.

## 5. Conclusions

A meta-analysis of prospective studies showed that there is insufficient evidence to show a difference in CRBSI rates between PICCs and tunneled catheters. On the other hand, PICCs showed less CRBSI rates than ports. Further, there was no difference in the rate of catheter-related thrombosis and mechanical complications. More prospective studies and randomized trials are needed to give specific recommendations for choosing between PICCs and tunneled central catheters to deliver HPN.

## Figures and Tables

**Figure 1 nutrients-11-02083-f001:**
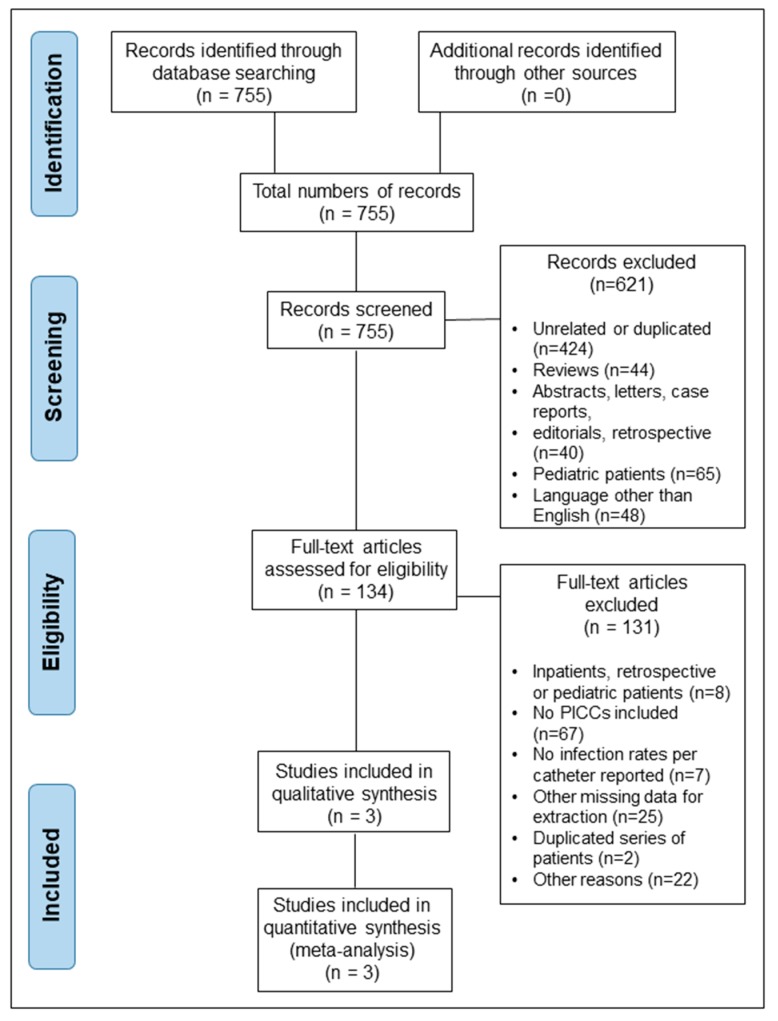
The Preferred Reporting Items for Systematic Reviews and Meta-Analyses (PRISMA) flow diagram.

**Figure 2 nutrients-11-02083-f002:**
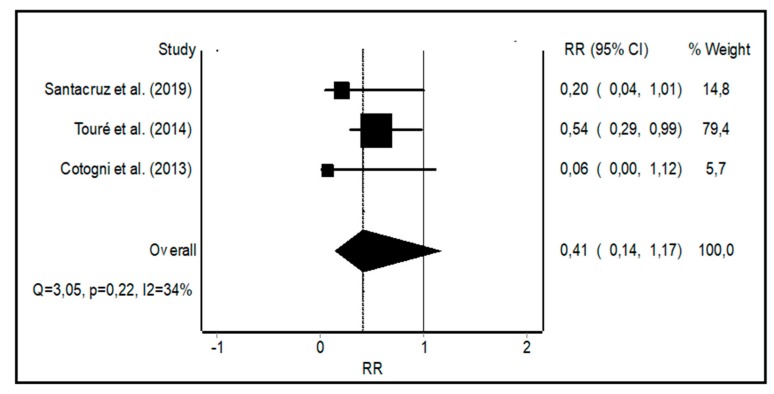
Forest plot showing the relative risk (RR) of catheter-related bloodstream infection (CRBSI) with peripherally inserted central catheters (PICCs) vs. tunneled central catheters. I^2^, a measure of heterogeneity.

**Figure 3 nutrients-11-02083-f003:**
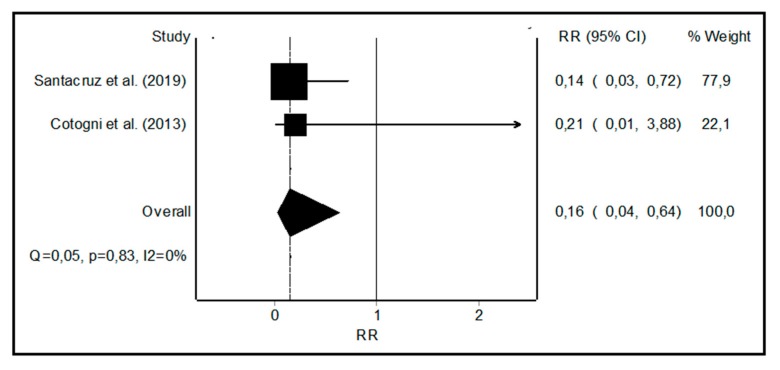
Forest plot showing the relative risk (RR) of catheter-related bloodstream infection (CRBSI) with peripherally inserted central catheters (PICCs) vs. ports. I^2^, a measure of heterogeneity.

**Figure 4 nutrients-11-02083-f004:**
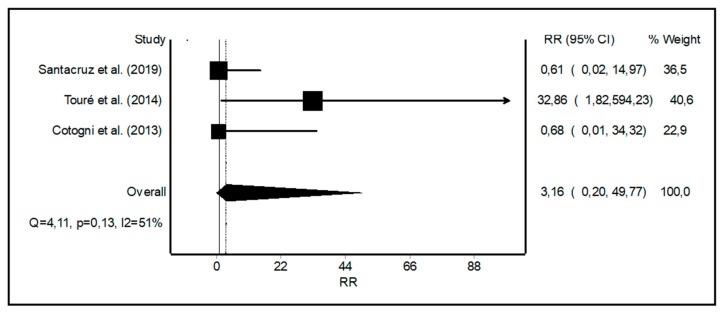
Forest plot showing the relative risk (RR) of catheter-related thrombosis with peripherally inserted central catheters (PICCs) vs. tunneled central catheters. *I*^2^, a measure of heterogeneity.

**Figure 5 nutrients-11-02083-f005:**
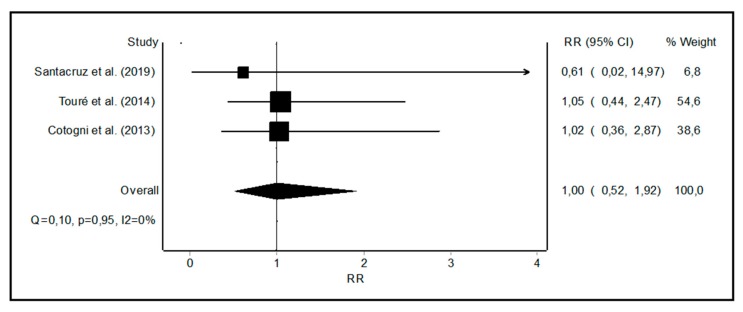
Forest plot showing the relative risk (RR) of catheter-related mechanical complications with peripherally inserted central catheters (PICCs) vs. tunneled central catheters. I^2^, a measure of heterogeneity.

**Table 1 nutrients-11-02083-t001:** Characteristics of the selected studies and patient demographics.

Author (Year)	Country	*N*	Mean Age (SD)	F/M	ON/Non-ON	Catheter Type	Catheter *n*
Santacruz et al. (2019) [7]	Spain	151	58 (13)	95/55	125/26	PICC Tunneled Ports	116 18 36
Touré et al. (2014) [18]	France	196	56 (17)	119/77	32/164	PICC Tunneled	71 133
Cotogni et al. (2013) [17]	Italy	254	67 (29–85) *	123/131	254/0	PICC Tunneled Ports Non-tunneled	65 45 72 107

*N*: number of patients, SD: standard deviation, F: females, M: males, ON: oncologic patients, * reported as median (interquartile range). PICC, peripherally inserted central catheter.

**Table 2 nutrients-11-02083-t002:** Catheter-related bloodstream infections and quality of studies.

Author (Year)	Catheter Days	CRBSI Rate *	CRBSI Diagnosis	Catheter Care	Catheter Lumens	Quality Index
Santacruz et al. (2019) [7]	PICC: 20495 Tunneled: 4167 Ports: 2970	0.15 0.72 2.35	Cultures **	Aseptic technique	ML:23 SL:101	12
Touré et al. (2014) [18]	PICC: 12322 Tunneled: 36812	1.05 1.87	Cultures **	NR	NR	11
Cotogni et al. (2013) [17]	PICC: 11504 Tunneled: 7835 Ports: 21605 Non-tunneled: 10364	0 0.64 0.19 0.87	Cultures **	Aseptic technique	All SL	11

NR: not reported; ML: multilumen; SL: single lumen.* Catheter-related bloodstream infections reported as episodes per 1000 days of catheter use. ** Cultures with a definition of catheter-related blood stream infection (CRBSI) as qualitative blood cultures from a peripheral vein and from the catheter, or which meet criteria for quantitative blood cultures or the differential time to positivity.

**Table 3 nutrients-11-02083-t003:** Catheter-related non-infectious complications in PICCs vs. tunneled catheters.

Author (Year)	Catheter Days	Catheter Insertion	Catheter Flushing	Thrombosis Rate *	Mechanical Complications *
Santacruz et al. (2019) [7]	PICC: 20495 Tunneled: 4167	US RG	Heparin	0.049 0	0.049 0
Touré et al. (2014) [18]	PICC: 12322 Tunneled: 36812	RG RG	Taurolidine-citrate ***	0.4 0	0.60 0.56
Cotogni et al. (2013) [17]	PICC: 11504 Tunneled: 7835	US US **	Saline	0 0	0.78 0.77

US: ultrasonography, RG: radiologic guidance. * Episodes per 1000 days of catheter use. ** or via blind venipuncture of the internal jugular or subclavian vein by specifically trained anesthesiologists and surgeons.*** taurolidine-citrate locks were injected in 35% of patients; for the rest, the flushing compound was not reported.

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
