# Peer review of "Infectious Complications in Home Parenteral Nutrition: A Systematic Review and Meta-Analysis Comparing Peripherally-Inserted Central Catheters with Other Central Catheters"

_nutrients, 2019, doi:10.3390/nu11092083_

Round 1
Reviewer 1 Report
The paper entitled “Infectious complications in home parenteral nutrition: a systematic review and meta-analysis comparing peripherally-inserted central catheters with other central catheters” compare only 3 prospective studies. The authors indicate in the limitations section, that concluding from 3 papers has low statistical power. I suggest changing the criteria of the study selection and enlarge studies comparing.
The following points should be explain:
why studies concerning pediatric patients were excluded? what was the “other reasons” of excluding of 22 studies?Author Response
Thank you very much for the opportunity to review our manuscript. We would also like to thank both Reviewers for their careful review and useful suggestions. We truly believe that the changes introduced in the manuscript following their indications have improved its quality. Here we submit the response to Reviewer 1:
1. The paper entitled “Infectious complications in home parenteral nutrition: a systematic review and meta-analysis comparing peripherally-inserted central catheters with other central catheters” compare only 3 prospective studies. The authors indicate in the limitations section, that concluding from 3 papers has low statistical power. I suggest changing the criteria of the study selection and enlarge studies comparing.
R. We are certainly aware of the low statistical power by including only three studies in the meta-analysis. However, as previously published systematic reviews on this topic did, by including also retrospective studies, the risk of bias is high, and the conclusions are even weaker. Furthermore, as explained in the Discussion, we think that by including only prospective studies, the readers will have the impression – as it is actually the case – that more prospective and randomized studies are needed as to definitely conclude which type of catheter is best for home parenteral nutrition.
2. The following points should be explain: why studies concerning pediatric patients were excluded?
R. We have no experience in pediatric patients, and home parenteral nutrition and the related diseases in this group of patients are distinct from adults. We have clarified this in the text (ln 104-105).
3. what was the “other reasons” of excluding of 22 studies?
R. We grouped those 22 studies under the heading “other reasons”. In fact, many of them had at the same time several of the exclusion criteria, such as being retrospective, missing data for extractions, or other different outcomes apart from catheter-related infections – use of taurolidine or locks for preventing infections – all together and therefore not categorized in the preceding headings. We have clarified this point in the text (Supplementary material pg 28 and ln 148-151 in the manuscript).
We hope the revised version is suitable for publication. Sincerely yours.
José I. Botella-Carretero MD, PhD, MBA
Reviewer 2 Report
Major critique
1) The authors state in their conclusions that PICCS have similar CRBSI rates than tunneled catheters. This is factually incorrect. We know that the relative risk of CRBSI rate was 0.41 (0.14–1.17) for PICC vs. 27 tunneled catheters. The null value of the confidence interval for the relative risk is 1. If a 95% CI for the relative risk includes the null value of 1, then there is insufficient evidence to conclude that the groups are statistically significantly different. It would be more accurate for the authors to say that there is insufficient evidence to show a statistical difference in CRBSI rates between the PICCS and tunneled catheter groups.
2) From line 338 the comment is made "Besides, the present results could not be applicable to other clinical settings in which a lack of specialized teams may compromise patients’ adequate training regarding the management of HPN and CVC care and identification of its possible complications". But we know that the use of home parenteral nutrition along with CVC care is carried out in specialised services. Therefore what exactly does this point say? Is it saying that the results here are not generalisable?
Minor critique
Some grammatical and syntax errors need to be rectified.
These include:
Line 43 - "intestinal transplant, the latter is therefore reserved for those presenting severe complications of parenteral nutrition"
This should be reworded as "presenting with severe.."
Line 46 - "the development of good education programs are a cornerstone to contribute to an improvement in the results"
This should be reworded as "a cornerstone in contributing to ..."
line 52 - "than peripherally-inserted central catheters (PICC) or tunneled-central catheters or single-lumen catheters in the North America"
An extra 'or' is unncessary and 'or' should be removed after (PICC) and a comma inserted instead
line 66 - "and in the view of the publication of some recent studies, we found interesting to perform"
This should be reworded as "we found it interesting.."
line 201 - "We were not able to meta-analyze the CRBSI rate comparing single lumen vs. multilumen catheters, as in one of the three studies only single lumen catheters were used [17], and in another one these data were not available [18] (Table 2)"
The term 'meta-analyze' is not a recognised term and should not be used here.
line 205 - "As PICCs and tunneled catheters showed no significant differences in the rate of infectious complications in the meta-analysis, but PICCs showed less infections than ports, as a secondary outcome of this meta-analysis we then compared the occurrence of other non-infectious complications between PICCs vs. tunneled catheters."
This is an unwieldly long sentence and should be rephrased
Author Response
Thank you very much for the opportunity to review our manuscript. We would also like to thank both Reviewers for their careful review and useful suggestions. We truly believe that the changes introduced in the manuscript following their indications have improved its quality. Here we submit the response to Reviewer 2:
Major critique
1) The authors state in their conclusions that PICCS have similar CRBSI rates than tunneled catheters. This is factually incorrect. We know that the relative risk of CRBSI rate was 0.41 (0.14–1.17) for PICC vs. 27 tunneled catheters. The null value of the confidence interval for the relative risk is 1. If a 95% CI for the relative risk includes the null value of 1, then there is insufficient evidence to conclude that the groups are statistically significantly different. It would be more accurate for the authors to say that there is insufficient evidence to show a statistical difference in CRBSI rates between the PICCS and tunneled catheter groups.
R. This is totally right, so we have changed the text as suggested by the Reviewer (ln 348-351) and also in the Abstract. Thanks for this important observation.
2) From line 338 the comment is made "Besides, the present results could not be applicable to other clinical settings in which a lack of specialized teams may compromise patients’ adequate training regarding the management of HPN and CVC care and identification of its possible complications". But we know that the use of home parenteral nutrition along with CVC care is carried out in specialised services. Therefore what exactly does this point say? Is it saying that the results here are not generalisable?
R. Again, the Reviewer is right. We have decided to remove this paragraph.
Minor critique
Some grammatical and syntax errors need to be rectified.
These include:
Line 43 - "intestinal transplant, the latter is therefore reserved for those presenting severe complications of parenteral nutrition"
This should be reworded as "presenting with severe.."
R. Corrected in the revised version of the manuscript. Thank you.
Line 46 - "the development of good education programs are a cornerstone to contribute to an improvement in the results"
This should be reworded as "a cornerstone in contributing to ..."
R. Corrected in the revised version of the manuscript. Thank you.
line 52 - "than peripherally-inserted central catheters (PICC) or tunneled-central catheters or single-lumen catheters in the North America"
An extra 'or' is unncessary and 'or' should be removed after (PICC) and a comma inserted instead
R. Corrected in the revised version of the manuscript. Thank you.
line 66 - "and in the view of the publication of some recent studies, we found interesting to perform"
This should be reworded as "we found it interesting.."
R. Corrected in the revised version of the manuscript. Thank you.
line 201 - "We were not able to meta-analyze the CRBSI rate comparing single lumen vs. multilumen catheters, as in one of the three studies only single lumen catheters were used [17], and in another one these data were not available [18] (Table 2)"
The term 'meta-analyze' is not a recognised term and should not be used here.
R. Corrected in the revised version of the manuscript. Thank you.
line 205 - "As PICCs and tunneled catheters showed no significant differences in the rate of infectious complications in the meta-analysis, but PICCs showed less infections than ports, as a secondary outcome of this meta-analysis we then compared the occurrence of other non-infectious complications between PICCs vs. tunneled catheters."
This is an unwieldly long sentence and should be rephrased
R. Corrected in the revised version of the manuscript. Thank you.
We hope the revised version is suitable for publication. Sincerely yours.
José I. Botella-Carretero MD, PhD, MBA
Round 2
Reviewer 1 Report
The paper has been improved and can be publish in the present form.
Reviewer 2 Report
Issues identified have been corrected.